# Improving the mental health of women intimate partner violence survivors: Findings from a realist review of psychosocial interventions

Sharli Anne Paphitis[1]*, Abigail Bentley[2¤], Laura Asher[3], David Osrin[4], Sian Oram[1]

1 Institute of Psychology, Psychiatry and Neuroscience, King's College London, London, United Kingdom, 2 Gender Violence and Health Centre, Department of Global Health and Development, London School of Hygiene and Tropical Medicine, London, United Kingdom, 3 Faculty of Medicine and Health Sciences, University of Nottingham, Nottingham, United Kingdom, 4 Institute for Global Health, University College London, London, United Kingdom

¤ Current address: Institute for Research in Social Welfare Politics (Polibienestar), University of Valencia, Valencia, Spain
* sharli.paphitis@kcl.ac.uk

**Data Availability Statement:** All relevant data are within the paper and its Supporting Information files.

## Abstract

### Background

Intimate partner violence (IPV) is highly prevalent and is associated with a range of mental health problems. A broad range of psychosocial interventions have been developed to support the recovery of women survivors of IPV, but their mechanisms of action remain unclear.

### Methods

Realist review following a prospectively published protocol in PROSPERO (CRD42018114207) and reported using the Realist and Meta-narrative Evidence Synthesis: Evolving Standards (RAMSES) guidelines.

### Results

Evidence was extracted from 60 reviews and triangulated in expert consultations. Mechanisms of action were categorised as either associated with intervention design and delivery or with specific intervention components (access to resources and services; safety, control and support; increased knowledge; alterations to affective states and cognitions; improved self-management; improved family and social relations).

### Conclusions

Findings suggest that psychosocial interventions to improve the mental health of women survivors of IPV have the greatest impact when they take a holistic view of the problem and provide individualised and trauma-informed support.

**Funding:** All authors on this study/project are funded in whole or part by the National Institute for Health Research (NIHR) Global Health Research Group (17/63/47). The views expressed are those of the authors and not necessarily those of the NIHR or the Department of Health and Social Care. The funders had no role in study design, data collection and analysis, decision to publish, or preparation of the manuscript.

**Competing interests:** The authors have declared that no competing interests exist.

# Introduction

Intimate Partner Violence (IPV) is one of the most prevalent forms of violence against women worldwide, suffered by 30% of ever-partnered women [1]. IPV is associated with a range of adverse health outcomes, including physical, sexual, and reproductive health problems and mental health problems such as depression, anxiety, substance abuse, and post-traumatic stress disorder [2–8].

Numerous psychological interventions have been developed to address mental health problems among survivors of IPV [8]. Additionally, a broad range of secondary and tertiary psychosocial IPV interventions report positive effects on mental health outcomes for IPV survivors. To date, however, reviews have not typically considered the underlying mechanisms of the broad range of psychosocial interventions, nor how their components might work together in complex interventions to improve mental health. This paper presents the findings of a realist review that aimed to explore the mechanisms by which psychosocial interventions improve mental health among survivors of IPV, the aspects of intervention design and delivery that influence whether the different mechanisms lead to an improvement in mental health, and the key knowledge gaps (both empirical and theoretical) that need to be addressed to enable more successful design and implementation of psychosocial interventions for survivors of IPV.

Realist syntheses examine how and why interventions may work in particular contexts or with particular populations [9]. From a realist perspective, causation within an intervention is grounded in its mechanisms of action. Following this line of reasoning, a realist synthesis considers 'families of mechanisms' in answering the question of what works, rather than 'families of interventions' as is often the case with systematic reviews and meta-analyses [9]. Developed from the idea that it is the underlying components (or "resources") within an intervention and the ways that individuals respond to them based on their circumstances that give rise to change, rather than the programmes themselves, a realist synthesis can begin to tell us how and why interventions may work in some contexts or for some populations [9]. Mechanisms of action from this realist perspective comprise both resources and reasonings. Resources (material, social, cognitive, or emotional) are provided by intervention programmes to participants, making new options and choices available to them [10]. Reasonings are the internal processes of the participants (be they survivors, perpetrators, service providers, or the wider community) that can contribute to change as a result of having access to resources. Synthesising evidence through a realist perspective can therefore produce Context-Mechanism-Output (CMO) statements that describe why and how components from different programmes work. These CMO statements become the unit of comparison that allow us to evaluate inconsistencies and develop programme theories to understand why and how interventions may or may not work for different populations and across different settings [9]. From the realist perspective, both the resources and reasoning components of the mechanisms of change in an intervention must be understood in order to capture the underlying causality at work [10].

Few realist syntheses of IPV interventions have been conducted, leaving a gap in knowledge around how and why interventions may or may not be successful. Previous realist reviews have included reviews of batterer intervention programmes [11], of IPV screening interventions in healthcare settings [12], and of advocacy interventions [13]. Our review adds to the literature on IPV interventions by providing a comprehensive examination of the mechanisms of action of multiple components across the full range of secondary and tertiary psychosocial IPV interventions. Its broad scope allows consideration of intervention components that may overlap and intersect, and the drawing out of families of similar mechanisms within and across IPV interventions that are important for initiating positive changes in survivor outcomes. We

focus specifically on the ways through which mental health outcomes can be targeted through various intervention components.

## Methods

The review was prospectively registered with PROSPERO (registration number CRD42018114207). We followed the RAMESES reporting standards [14] and adhered to Pawson's methodology for realist synthesis, iteratively conducting background searches, searching for programme theory, searching for empirical evidence, and refining programme theories [15, 16].

### Exploratory scoping

Ten review articles were selected by a group of subject experts from our NIHR Global Health Research Group (S1 File) to inform initial exploratory scoping. To produce a preliminary conceptual model to guide the review (S2 File), elements of programme theory were extracted from these 10 articles and used in conjunction with a Theory of Change designed by the NIHR Global Health Research Group to inform the development of a package of care for survivors of violence in South Asia (https://fundingawards.nihr.ac.uk/award/17/63/47). This preliminary conceptual model presented an overview of how and why changes as a result of interventions investigated in the review were expected to happen.

### Searching, selection and appraisal process

Scoping and initial mapping identified firstly that there was a broad range of interventions relevant to building multi-component programme theory; and secondly that there were a number of review articles covering the broader theory, interventions, and components. We therefore decided to focus our search and CMO development on the evidence available at the review level. Synthesising evidence at review level facilitated the broad scope of our study and allowed us to consider mechanisms across a wide variety of domains. The use of reviews also served to answer the question 'what works', given that the majority of reviews reported on the effectiveness of the interventions they were evaluating along with an assessment of study quality. This allowed us to direct more attention towards the questions of how and why interventions work, questions that have been underexplored in the IPV literature. We conducted searches for articles in PubMed, SCOPUS, Cochrane and Open Grey, as well as a Google Scholar search and hand searches in relevant journals to expand the exploratory scoping. We used the broadest possible search terms: "intervention", "intimate partner violence", "domestic violence", "review". Searches were limited to articles in English from January 2000 - January 2020.

Titles and abstracts were screened by one reviewer. A second reviewer independently reviewed a random sample of 250 of titles and abstracts. At each stage, disagreement was resolved by discussion between the two reviewers: reference to a third reviewer for consensus was not required. Papers included at the full-text stage were screened independently by two reviewers. Inclusion criteria required papers to be review articles on psychosocial interventions for women survivors of IPV. Psychosocial interventions were defined in the protocol as activities, techniques, or strategies that target biological, behavioural, cognitive, emotional, interpersonal, social, or environmental factors with the broad aim of improving health, functioning, and wellbeing.

Searches yielded 8360 articles and removal of duplicates and title and abstract screening left 125 review articles for full-text review (Fig 1). 60 reviews were retained after full-text review and contributed to the synthesis (S3 File). No materials were excluded based on overall judgement of their risk of bias, in line with realist thinking [14]. The relevance of materials was the

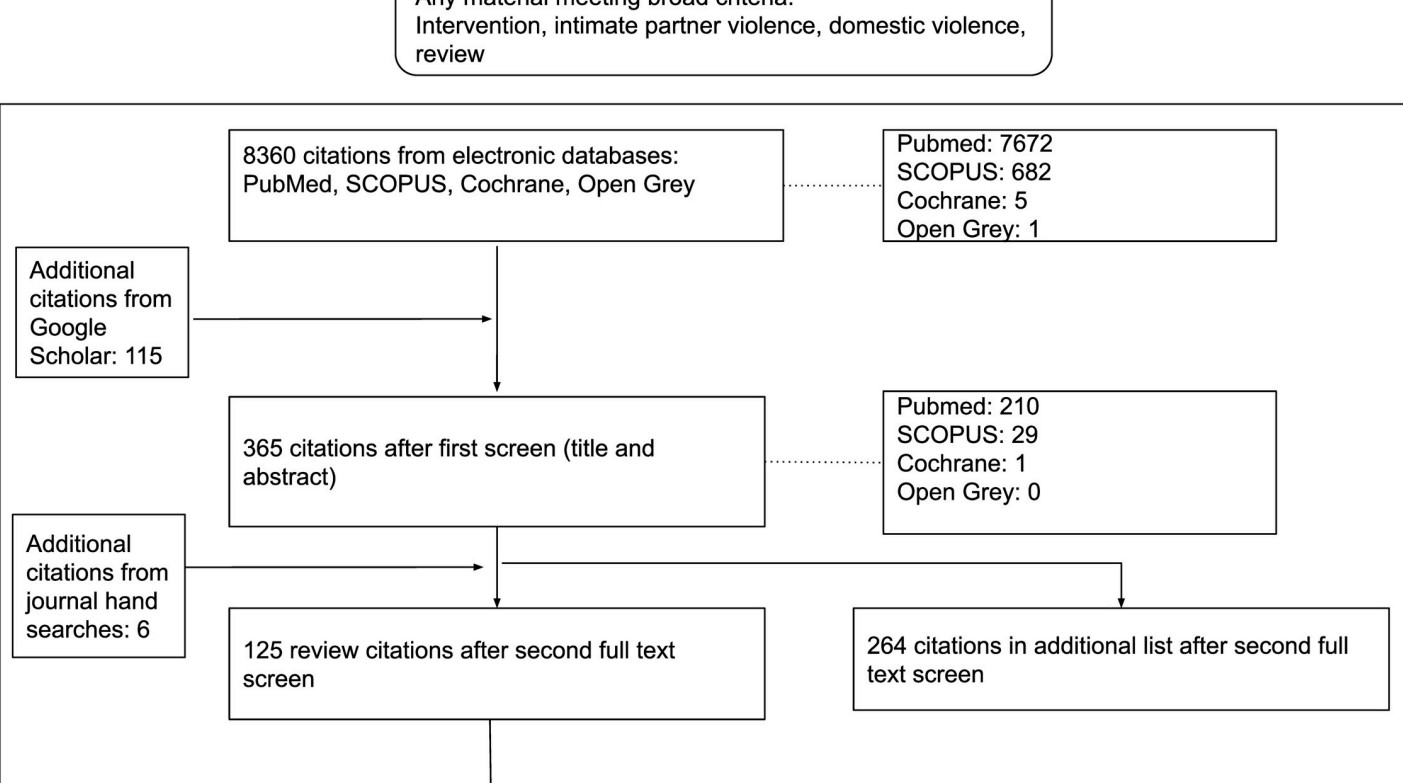

**Fig 1. Flow diagram illustrating the search process and article disposition.**

primary criterion for inclusion in the synthesis; relevance here defined as the likely contribution to the development, refinement, and testing of theory.

## Data extraction

Information extracted into a standardised MS Excel sheet included bibliographic information, countries and settings, sample characteristics, types of interventions, theoretical considerations, programme theories, intervention activities, design and delivery of interventions, participant experiences, provider experiences, community responses and experiences, outcome measures, and results. Text relevant to the context, mechanisms, delivery, and outcomes of interventions was then extracted into a linked MS Excel Sheet for further analysis and synthesis to develop the final conceptual model and CMO statements.

## Analysis and synthesis process

Text relevant to the intervention resources and reasonings, mechanisms of change, intervention contexts, and outcomes was coded independently by two reviewers using a thematic

approach. The thematic coding was structured around the type of mechanisms and underlying theories explaining the pathways from interventions to outcomes, aiming to test and refine the conceptual model developed through initial scoping and to produce middle range-theories. The initial analysis sought to identify emerging patterns ("demi-regularities") within the overall families of mechanisms highlighted in the preliminary conceptual model. Candidate middle-range theories comprising connections between resources and reasonings to explain the emerging patterns were discussed amongst the research team. Emerging CMO configurations were used to redraft the conceptual model throughout an iterative analysis process.

272 CMO configurations were developed and categorised into families of mechanisms of action as outlined in the conceptual model. Two 3-hour stakeholder engagement workshops were subsequently held with invited IPV intervention experts, mental health practitioners, service provider staff, and intervention programme facilitators based at two of our partner non-governmental organisations in India: SNEHA and Sangath. Workshops were facilitated by the review team and discussed families of mechanisms and CMO configurations. During the workshops, qualitative input was collected on the overall relevance and plausibility of the proposed mechanisms in intervention settings.

The long list of 272 CMO configurations was condensed through a final round of thematic analysis into 73 final CMO statements. Criteria were developed to determine the relative weight each statement in the long list of CMO configurations would be given in the final model, based on the following criteria:

1. Number of reviews with evidence supporting the statement.

2. Whether the evidence in support of the statement from the review was primarily data or theory driven.

3. Whether the evidence from the review in support of the statement was 'thick' or 'thin'.

4. Whether the statement had been confirmed by the qualitative data collected during the workshops.

5. Overall coherence of the statement within the broader mechanism.

With respect to criterion 2, we did not prioritise either theoretical or data driven insights, instead seeing statements with a wide diversity of insight as well supported. In evaluating criterion 3, CMO statements were coded based on whether they had come from 'thicker' descriptions of underlying mechanisms directly from the reviews, or whether descriptions of mechanisms were 'thinner' and had less detail, thereby requiring some qualitative extrapolation by us. Ultimately, CMO statements that were constructed from multiple reviews, were both theory and data driven, were supported by thick evidence, and were validated during the expert workshops were viewed as the most well supported within the current literature base.

## Results

The final conceptual model (Fig 2) summarises the contexts, mechanisms, and outcomes identified across a broad range of psychosocial interventions as supporting the recovery of women survivors of IPV. Mechanisms have been broadly split into two sections: those detailing the design and delivery elements of interventions, and those detailing intervention component elements. Tables accompanying each of the mechanisms have been constructed from the CMO statements and attempt to depict a logical progression of mechanisms of action leading to outcomes (S4 File). In reality, there are likely to be multiple feedback loops within and between mechanisms, as well as between mechanisms and outcomes, given that complex interventions

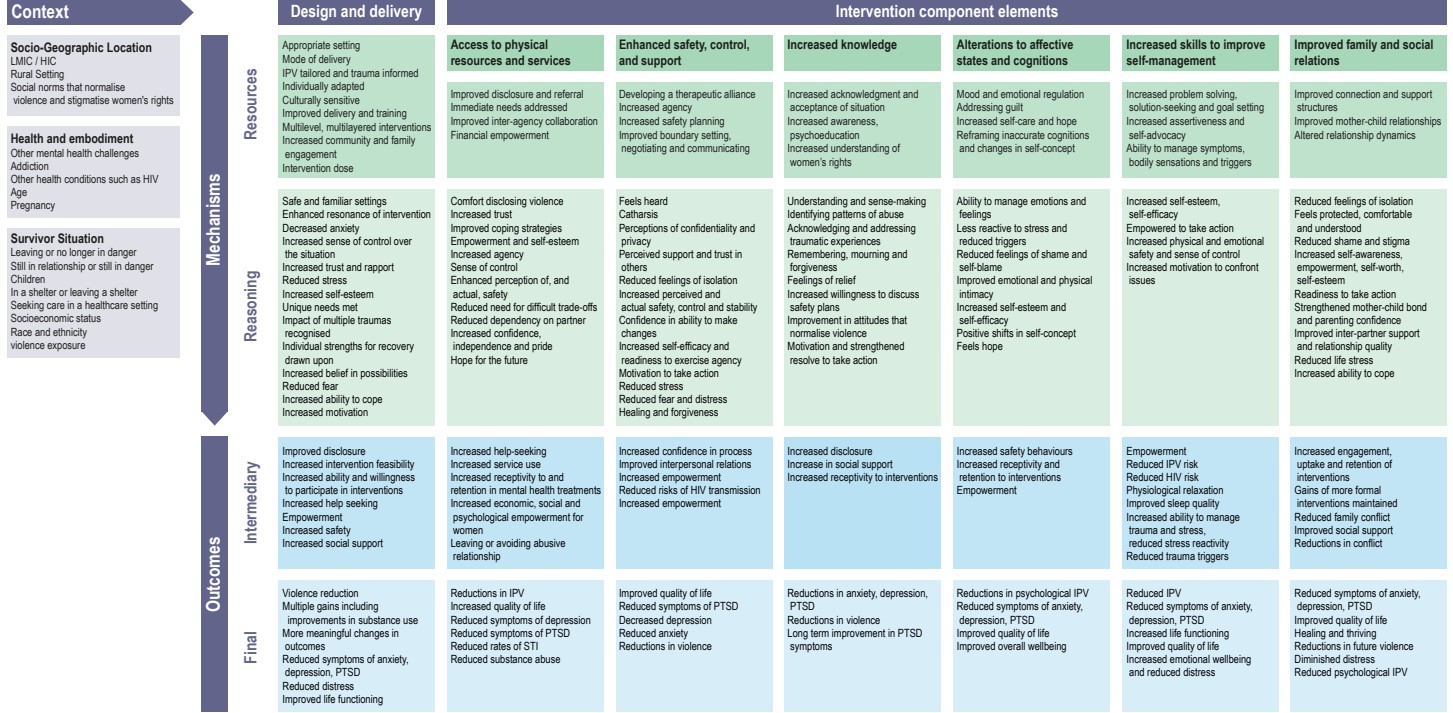

**Fig 2. Final version of the conceptual model, mapping intervention components, outcomes, and mechanisms.**

"are open systems that feed back on themselves" [17] and shifts in one constituent are likely to set in motion additional mechanisms that lead to additional changes.

## Design and delivery

The 'where, how, who, and how-long' of intervention design impacts on feasibility, acceptability, and buy-in for interventions, which in turn affect their outcomes.

**Where.**  Intervention feasibility and acceptability, as well as perceptions of feasibility, play an important role for outcomes. Whilst interventions in settings that are accessible and familiar to survivors ensure that low-income and marginalised populations can be reached, a lack of privacy and safety in intervention settings can increase anxiety for survivors and lead to decreased uptake and retention [18–29].

**How.**  Interventions that are individually adapted, IPV-tailored, and trauma-informed are likely to yield the best results. IPV-adapted trauma treatments that include strategies to manage trauma symptoms that can be applied to the specific and continuing stressor of IPV are particularly helpful for women who face continuing abuse or who are in shelter settings, leading to an increase in their sense of control over the situation and reducing symptoms of anxiety [19, 20, 27, 28, 30–33]. Tailoring interventions to meet the unique and evolving needs of individual survivors can lead to more meaningful changes in outcomes since women may be at different stages of readiness to address their situation and programmes targeting a predefined set of outcomes (such as depression or PTSD) may not meet their unique needs or draw on their individual strengths for recovery [13, 19, 27, 28, 30, 34].

Participant and wider community buy-in and acceptance of interventions are important for uptake and retention. Culturally sensitive interventions that include a deep understanding of the contexts in which IPV takes place, community understandings and experiences of mental

health, and help seeking and coping behaviours, can improve survivor trust, uptake, and engagement in interventions [13, 20, 24, 30, 35–38]. When interventions are relevant for and resonate with communities, communities are empowered to address some of the underlying causes of violence. Engaged approaches that foster community partnerships and use participatory approaches to generate community-wide solutions can improve the likelihood of survivors being able to seek help by ensuring programmes can continue to run, and can reduce the risk of violence and survivor distress by empowering communities to support survivors and address the perpetration of violence [18, 22, 23, 35, 36, 39–42]. However, interventions that are not informed by community engagement strategies, norms, and cultural contexts could prevent survivors' access to appropriate support due to fears of community retaliation, and increase their anxiety or their risk of harm. This could erode their trust in formal services and reduce help-seeking [13, 18, 23, 24, 30, 35, 36, 38, 40, 41, 43, 44]. The prevalence of harmful stereotypes about survivors from particular cultural or socio-economic groups can undermine the ability of women to participate in interventions and treatments. Interventions that are not contextually and culturally adapted to include an understanding of these stereotypes and specific related barriers will have poor uptake, retention, and outcomes [38].

**Who.**   Increased acceptability and feasibility of interventions for service providers, which can often be achieved through integrating IPV interventions into existing services, can influence provider buy-in and subsequent delivery and implementation [27, 28, 45–47]. The provision of adequate and ongoing provider training by organisations and institutions, particularly with mental health content, can increase survivor disclosure and the effectiveness of interventions due to increased provider engagement [20, 21, 28, 29, 46–49]. However, a lack of physical, financial, or human resources, clear protocols, and adequate networks at the institutional level can lead to reduced provider buy-in and willingness to implement interventions due to increased frustration at not being able to provide the necessary services and concerns about lack of skills [13, 18, 21, 26, 27, 29, 35, 43, 45, 48, 50–52].

Interventions that are multi-layered and cut across a range of underlying problems can contribute to improved outcomes by utilising pre-existing trusting relationships with other service providers and can lead to multiple gains across a range of outcomes [20, 26, 28, 30, 32, 53–56]. A lack of multi-layered interventions can make services inaccessible for women who have multiple and complex needs, such as those with substance abuse problems and complex mental health problems. These women are often excluded from interventions and, as such, the underlying causes of their difficulties often go unrecognised, leading to inappropriate treatments and unsuccessful referrals. However, multilevel interventions are resource intensive and may not be feasible in resource-constrained or busy primary care settings [27, 28, 30, 56].

**How long.**   Whilst brief and short-term interventions that extend beyond a single session can be effective in busy primary care and shelter settings [19, 26, 30, 45, 47, 48, 50, 57], increased treatment frequency and a greater amount of time spent in treatment, particularly 10 sessions or more, can lead to greater improvements in mental health and reductions in violence due to the increased amount of time to address the complex nature of IPV and multiple needs of survivors [19, 28, 32, 57]. Brief therapies with well-coordinated follow up or booster sessions may be particularly beneficial for women while based in shelter settings [19, 30].

## Intervention component elements

Families of mechanisms within the intervention component section include (i) access to resources and services; (ii) enhanced safety, control, and support; (iii) increased knowledge; (iv) alterations to affective states and cognitions; (v) increased skills to improve self-management; and (vi) improved family and social relations.

**Access to resources and services.** This family of mechanisms operate by reshaping the survivor's World-Self relationship, supporting them in regaining a sense of control over the ways in which the external world and the vicissitudes of fortune impact on the self. Reshaping the World-Self relationship takes place through increased access to resources and the availability of viable life options to address the survivor's situation, which leads to increased agency and self-efficacy. Increasing a woman's access to resources supports her to address her immediate needs and financial independence, as well as facilitating access to services via effective screening, referrals, and interagency collaborations. Ensuring that women have access to the resources and services that can benefit them also means that they can focus on healing from their experiences rather than having to navigate complex situations and systems that may detract from their recovery [13, 19, 28, 30, 33, 43, 44, 55].

Access to resources and services may be mediated by disclosure of IPV and onward referral. An effective system that incorporates enquiry with immediate onsite referrals and referrals to outside organisations providing IPV services, alongside adequate provider training and supportive institutional policies, can motivate providers to make and survivors to seek referral. Uncoordinated systems and a lack of integrated referrals result in women having to navigate systems alone and seek help from multiple independent services. This leads to frustration, fatigue, and a reduced desire to seek help. Whilst the integration of services to create multi-sectoral programmes can have promising outcomes, a lack of training, skills, and infrastructure can create large variations in quality and may not be sustainable in all settings [20, 43, 44, 51, 52, 54]. IPV advocates working with and on behalf of survivors can, however, help women access a range of services and navigate relevant systems [13, 30, 43, 46].

Women's financial independence can be addressed by economic interventions, particularly those that include social components that engage men around gender norms and equity. This can increase women's social capital and empowerment and lead to a reduction in IPV risk and improved mental and physical health outcomes [23, 32, 53–55, 58]. However, interventions that increase a woman's economic empowerment or independence, particularly when not combined with social intervention components to address elements such as safety planning, can increase the risk of violence in the home [13, 30, 32, 49, 53, 58].

**Enhanced safety, control and support.** This family of mechanisms operate by reshaping Other-Self relations, allowing survivors to establish a level of self-protection and regain a sense of control over the ways in which others' actions and relations with others affect them and shape their lives. Reshaping Other-Self relations takes place through supportive and collaborative efforts that empower survivors to maximise their safety, leading to increases in their agency and self-efficacy.

The prioritisation of survivor safety is crucial. Increased safety may be achieved through the development of a therapeutic alliance between survivor and service-provider, but also through concerted efforts to plan actively for safety, set boundaries, and negotiate and communicate effectively. Disclosure of trauma and development of a therapeutic alliance can itself play a therapeutic role, allowing survivors to feel listened to and reducing feelings of isolation [13, 18, 21, 30, 45, 47, 51, 56, 59]. The therapeutic alliance – and women's empowerment – can be strengthened by enabling survivors to exercise their own agency in the recovery process by encouraging them to talk through solutions, discover their own goals, and make their own choices [13, 26, 33, 51, 60–62].

Interventions that include safety planning can equip survivors with knowledge and skills, empowering and motivating them to act, particularly when they may fear for their safety or are actively planning to leave an abusive relationship [22, 24, 26, 32–34, 37, 63]. Safety plans should consider survivors' exposure to different types of violence as well as their specific needs. However, safety planning alone may not be sufficient to help increase women's sense of

security and promote recovery [24, 25, 28, 43, 44]. Safety is also moderated by the woman's level of dependency on her partner and the abuser's influence on her through a continuing relationship or ongoing contact, including because of children, with implications for mental health, coping, and accessing treatment and support [13, 29, 30, 33, 61].

Increased agency and control for survivors, such as through increased capacity to make choices throughout the intervention and recovery process, can contribute to improvements in safety and self-efficacy, leading to improvements in mental health [13, 26, 33, 51, 60–62]. Interventions that assist survivors with strategies and skills to manage contact with perpetrators—such as boundary setting, negotiating, and communicating—can lead to increased actual or perceived safety and control and reduce fear and distress. The use of restorative justice approaches to mediate contact between survivor and perpetrator could also help women to feel heard, allowing them to work towards healing and forgiveness [30, 36].

**Increased knowledge.** This family of mechanisms operate through growth in the survivor's Epistemic-Self: expanding what survivors know and understand about abuse and available resources, reframing how they interpret and make meaning of the world and their experiences, and increasing their ability to use this knowledge and understanding for personal sense-making, self-advocacy, and healing, thereby increasing their epistemic agency.

Survivors' acknowledgement of their situations and acceptance that what they are experiencing is violence, knowledge about the causes and consequences of IPV, and knowledge of available resources can enable them to identify patterns of abuse and motivate them to seek help and take action to support their recovery, including by discussing safety plans with providers. Increased knowledge of the causes and consequences of IPV may increase receptivity to interventions and improve psychological symptoms [23, 24, 30, 33, 54, 55, 61]. Coming to understand and make personal sense of violence can help women reach a place of acknowledgement, acceptance, and sometimes even forgiveness, which can play an important role in their recovery.

Educational activities that foster dialogue and raise awareness in communities and between couples can increase understanding of women's rights and lead to shifts in gender equitable norms and beliefs and the acceptability of IPV, resulting in improved social support for survivors and a reduction in IPV [18, 41, 53, 61]. However, women's ability to recognise and understand violent behaviours and abuse as problematic can be influenced by social and cultural beliefs. This can lead to a lack of trust in, internal conflict over, and confusion from educational and awareness raising activities that do not align with these views and beliefs [33].

**Alterations to affective states and cognitions.** This family of mechanisms operate by reshaping survivors' Self-Self relations, allowing survivors to manage their internal states and reframe misconceptions about themselves, leading to increased self-esteem and hope. Reshaping Self-Self relations takes place through a variety of activities centred on developing and supporting women's inner resources. These include learning skills to control emotions, coming to understand and heal from traumatic experiences, learning ways to understand and overcome intruding thoughts, guilt, and shame, and modifying and reversing a range of unhelpful beliefs, misconceptions, and patterns of thinking.

Encouraging reconnection between self and body through the normalisation of body talk and the improvement of self-care activities can improve empowerment and self-efficacy—increasing women's ability to feel hope—and emotional and physical intimacy, leading to improvements in psychological outcomes [30, 37, 57, 64].

Even when offered for brief periods and for women who are still exposed to ongoing abuse, interventions that focus on cognitive reframing and thought stopping skills can assist survivors in assessing and reframing negative beliefs and inaccurate cognitions that maintain trauma symptoms, bringing about positive shifts in self-concept [30, 39, 43, 44, 61, 62]. However,

interventions focused on shifting cognitions and affective states are unlikely to be effective if they do not also ensure that women are supported in gaining access to the physical and social resources they need to mitigate the effects of traumatic stress caused by resource loss, which undermines survivor coping, increases dependency on abusers, exacerbates PTSD symptoms, and increases psychological distress [19, 44].

**Increased skills to improve self-management.** This family of mechanisms operate by reshaping and enhancing survivors' Self-World relations, allowing them to manage and regain control over internal states that shape their actions and behaviours in the world, and internal states that arise in response to stimuli from the external world. This supports survivors in cultivating a belief in their own ability to take actions that are in their own best interest.

Building skills that improve mind-body relations and bodily functioning can support survivors in identifying and coping with stress or events that may 'trigger' the same physiological or emotional reactions experienced during abuse and allow them to address physiological factors that can compromise their ability to engage in new behaviours. This can allow survivors to regain a sense of control and improve mental health outcomes [30, 61, 64].

Enhancing assertiveness and self-advocacy skills helps survivors to replace counterproductive and emotionally-driven behaviours through improved self-esteem, allowing them to take concrete actions to reduce risk, particularly when faced with the likelihood of ongoing contact with their abuser and a high risk of revictimisation. This can lead to improvements in emotional wellbeing and mental health [19, 30, 55]. Enhancing problem-solving and solution-seeking skills supports survivors in making decisions and taking effective action by increasing self-esteem, sense of control, and motivation to confront issues, and can lead to improvements in quality of life and mental health [19, 26–28, 30, 31, 37, 40, 43, 46, 49, 60, 61, 63, 65]. However, for women experiencing high levels of physical violence who may look to others for solutions, inadequate structures and systems in service provision that prevent providers from presenting clear or effective solutions can undermine the solution-seeking components of interventions that aim to build self-efficacy and empowerment [21, 33, 45].

**Improved family and social relations.** This family of mechanisms operate by enhancing survivors' Self-Other relations, allowing them to manage and improve their interpersonal relationships and broaden their base of social support by building positive connections with other survivors, friendship groups, families, or communities.

Improvements in social connectedness allow survivors to feel supported, increasing self-esteem and feelings of self-worth and leading to improvements in their willingness and ability to take action as well as their ability to cope. Strengthening informal social support structures through talking circles, group therapy, and support groups can allow survivors to feel comforted, protected, and understood. Improved social support can also facilitate sharing of information to improve help-seeking and safety behaviours and can mobilise social networks to eliminate shame and stigma around abuse, all of which can aid in the recovery process [20, 24, 30, 33, 35, 36, 40, 44, 59, 66]. This is particularly important for older or marginalised women with limited social networks [38, 67].

Relationships between abused women and their children may be improved through mother and child interventions that enhance parenting skills and feelings of empathy, with the potential for diminished distress for survivors, improved child behaviour, and improved psychological outcomes for survivors and their children [24, 27, 28, 30, 39, 65]. Joint interventions that view the child as an agent of change can foster recovery in both parties through strengthening the mother-child bond [39, 65]. However, resource constraints for the provider or survivor, or the deliberate use of children in abuse perpetration patterns, could undermine the process of mother-child interventions by interfering with the mother-child relationship and reducing access to interventions [28, 53, 65].

Relationships between women and their partners may be improved through joint couple interventions that encourage positive relationship behaviours, foster intra-partner support, build communication skills between partners, and provide women with the skills to re-evaluate their relationships. Such interventions can lead to alterations in relationship dynamics through women's increased self-efficacy, readiness to take action, and safety behaviours. This can reduce conflict, improve mental health, and lead to reductions in IPV risk, particularly for couples who do not wish to separate [24, 27, 32, 53, 62, 68–71].

When services are contingent on women leaving the relationship or promote the idea that perpetrators need to be isolated from survivors, couples can be prevented from seeking help and this may lead to progression of violence [68]. However, couples-based interventions may not be appropriate when violence is largely motivated by coercive control and domination or when there is a high risk of ongoing physical violence or substance abuse issues as they may cause iatrogenic harm and compromise safety [71]. Couples therapy could also lead to victim-blaming and feelings of intimidation in survivors, and could help perpetrators find new ways to influence, control, and coerce their partners [53, 71].

## Discussion

### A realist model of IPV interventions

Our realist review sought to identify the mechanisms of action by which psychosocial interventions improve the mental health of women survivors of IPV. Six families of mechanisms emerged which, taken together, provide a coherent and comprehensive model describing how IPV interventions can help survivors move towards recovery and improved mental health (Fig 2). The model is grounded in design and delivery mechanisms that increase the feasibility, acceptability, and effectiveness of the intervention components. Mechanisms were common across multiple intervention types and many interventions incorporated multiple mechanisms. We suggest that mechanisms, rather than the specific intervention type, should be the key focus for the future development of IPV psychosocial programmes.

Given the inclusive nature of the review, our findings also identified a range of intermediary and final outcomes in addition to mental health that are important for women's improved wellbeing, recovery, and healing. These include reduced risk of violence, reductions in family conflict, improved interpersonal relations, economic, social, and psychological empowerment, improvements in self-concept, and strengthening of social support. In line with one of the central messages of our study, that a holistic approach to intervention and recovery is vital, this allowed us to present a comprehensive overview of the interventions, resources, reasonings, and outcomes contained within the current IPV intervention literature.

Our model's descriptions of the mechanisms through which interventions affect change (the how and the why) is supported by existing work detailing the common components of domestic violence programmes and models of trauma recovery. For example, key features of domestic violence programmes identified by the Domestic Violence Evidence Project [72] align with the mechanisms in our model, suggesting that they are core features of programmes that aim to improve survivors' mental health. Herman's three-stage model of trauma-recovery [73], widely cited in the IPV literature, closely reflects our model's mechanisms of action contributing to cognitive, behavioural, and emotional changes; as do the "intrapersonal changes" described by Sullivan's social and emotional well-being conceptual framework of domestic violence interventions [72]. Although focused on secondary and tertiary prevention programmes, our model is further supported by frameworks for primary prevention and by models developed with different populations and settings. The WHO RESPECT framework includes seven

strategies to prevent violence against women [74]. All seven have overlaps with our model's mechanisms.

The overlap of strategies between our model and primary prevention frameworks speaks to the merits of an integrated and holistic model of care for survivors of violence. The model is not divided into a linear sequence or stages of recovery, but instead presents a collection of mechanisms that evidence suggests ought to be present within IPV interventions. Although we present families of discrete mechanisms, we do not suggest that families or individual mechanisms should be addressed in isolation. Ideally, IPV interventions should strive to incorporate as many mechanisms as are possible, the grouping, order, and structure being guided by contextual considerations. Where practical challenges mean this is not possible, meaningful working partnerships should be put in place and resources should be allocated to ensure that survivors are adequately supported through referral processes.

Our model provides an overview of the mechanisms that can be, and have been, included in IPV interventions throughout the global literature. Discussing all possible contexts was beyond the scope of the review and the model should therefore be used as a starting point requiring further work to contextualise it: considering specific geographical and cultural settings, resource rich or poor settings, specificity of violence situations and differing perpetration patterns, and considerations about whether women predominantly want to stay in their relationships, want to leave, or have already left.

## Mechanisms and components

**The significance of design and delivery.**   Design and delivery mechanisms and specific intervention component mechanisms are intimately connected and affect one another. Regardless of intervention type or specific psychological therapy, intervention development should include careful consideration of design and delivery elements, and the impact of resources and reasonings contained within them, in addition to how they intersect with the mechanisms of specific intervention components.

In particular, interventions are likely to yield the best results where they are perceived as feasible by practitioners and survivors and are culturally-relevant, individually adapted, IPV tailored, and trauma-informed. When interventions provide psychological therapies, but place limited focus on the wider structural support necessary to introduce new programmes, particularly when institutional or community support is lacking, or when they are not culturally or contextually relevant, they are less successful. Triggering successful mechanisms requires intervention resources to be relevant and acceptable to both survivors and providers: neither group should be overlooked during intervention development. Ensuring that providers feel adequately supported in their roles will improve their acceptance and delivery of interventions, and subsequently improve survivor outcomes [13, 18, 21, 26, 27, 29, 35, 43, 45, 48, 50, 51, 56].

The need for individualised and IPV-tailored interventions was a prominent theme throughout the review, affecting all of the mechanisms we identified. Achieving this requires that interventions take account of the complex nature of IPV and women's particular experiences of abuse, including sexual and emotional abuse. Given that violence patterns are complex and nuanced, an understanding of women's specific exposure history is an essential aspect of providing the most effective immediate and ongoing support. Surprisingly, the need to understand exposure histories and tailor services and interventions accordingly is not particularly prominent in the literature, and further research is needed to understand the relationships between particular types of victimisation and the associated mental health outcomes, as well as how to effectively tailor responses [75]. Individualisation should also incorporate an understanding of the context of abuse, in order to direct survivors - and perpetrators - to the

most appropriate resources. Where IPV perpetration is characterised by coercive control, psychoeducation, awareness-raising, and elements of primary prevention may be the safest and most appropriate methods of intervention for couples. Alternatively, if violence is used as a tool for conflict resolution and emerges as a result of environmental stressors, meeting basic needs and providing couples with tools to manage stress and improve communication may be effective [71]. Given all of this, we suggest that the development and use of nuanced assessment tools could highlight both patterns of violence and women's particular mental health priorities and needs (including the identification of more severe and complex mental health concerns which may require specialist treatment).

As well as increasing the likelihood of intervention acceptability, uptake, and retention, the design and delivery aspects of the interventions themselves have a significant bearing on intermediary and final mental health outcomes. For example, by holding sessions in private and secure spaces, with short waiting times for appointments, and in locations relevant to women's lives such as churches or schools, partner suspicion can be reduced and women's sense of safety increased, leading to reduction in anxiety [24, 29].

**Critical components.** One of the critical issues highlighted by our review is the way in which IPV affects every facet of a woman's life. Survivors experience multiple and competing negative psychosocial concerns, including the protection and care of children, physical health concerns, safety concerns, financial instability, legal proceedings, feelings of isolation and lack of social support, low self-esteem and feelings of grief, and managing ongoing threats from and relations with abusers, as well as ongoing trauma and psychological symptoms. Unless the myriad of pressing psychosocial and practical concerns weighing on survivors are addressed, it is unlikely that they can be fully supported in their journey to recovery [28, 30]. A woman's safety and stabilising her situation should be a priority for any intervention before steps are taken to provide more targeted therapeutic care. This is where immediate crisis intervention procedures should be considered.

The findings of our review demonstrate that a wide variety of intervention components from across a range of psychosocial interventions can trigger mechanisms that are advantageous for survivor mental health. For example, interventions that improve women's social support, that teach them about problem solving, goal setting, boundary setting, and negotiating, that help them to manage stress through breathing techniques, muscle relaxation, and mindfulness, or that encourage the development of a therapeutic alliance with a provider can lead to improvements in mental health. In situations where resources do allow targeted psychological therapies, the addition of complementary intervention components that can also directly or indirectly affect mental health outcomes could be advantageous for prolonging gains or increasing the likelihood of success. An illustrative example from our synthesis is that interventions that seek to address survivors' basic needs alongside providing trauma-focused CBT (TF-CBT) may increase the effectiveness of TF-CBT by helping women to be more receptive to, and engaged with, mental health treatment once their immediate concerns have been acknowledged [44, 62].

In responding to stressful life experiences people employ coping strategies that can be either maladaptive, such as avoidance or substance abuse, or adaptive, including help-seeking and marshalling internal and external support and resources to address the problems that lead to increasing stress [75, 76]. Central, and interrelated, factors supporting survivors' adaptive coping strategies are self-efficacy, agency, and empowerment, all of which play a role in shaping beliefs and attitudes (internal reasoning) about their ability to effectively plan and manage change in their situation [75, 77, 78]. Victimisation is associated with lower levels of self-efficacy, eroding survivors' positive internal beliefs about the relationship between the self, the world, and the other, while the development of self-efficacy is nurtured through supportive

and affirming interpersonal relations [75]. Consistent with this, our findings suggest that connection is an important route to healing for survivors of IPV, regardless of whether it is with family, friends, religious or community leaders, other survivors, service providers, or children. Components that developed or strengthened connection were apparent throughout the model as mechanisms that improve mental health. These include the importance of building a therapeutic alliance, the importance of fostering perceived or actual social support, and the effectiveness of group and mother-child interventions.

Our review also highlighted that a lack of multi-layered approaches can exclude some of the most vulnerable women from accessing relevant services, particularly those with complex mental health needs [30, 56]. When survivors of IPV are unable to access appropriate services to address all of their needs, they may be deterred from seeking help. Research and interventions tend to exclude women with severe mental health conditions [30], contributing to a lack of understanding of how to address complex mental health needs in survivors of IPV.

All of this suggests that an effective way to work with survivors of IPV would be to use multi-layered approaches and to combine multiple components into an overarching intervention that aims to operate through multiple mechanisms. Multilevel and multi-layered interventions that are able to address the variety of challenges faced by survivors can lead to increased access to resources, reduced violence, and improved mental health. Complex interventions for survivors of IPV are increasingly being adopted [79–81], although there still appears to be a lack of integration between the violence and mental health disciplines that prevents multi-layered approaches from being mainstreamed and survivors from accessing resources that give them the best chance of recovery.

Women in abusive relationships may not always want to leave their partners. They may instead be interested in finding solutions to reduce violence, increase their autonomy, and improve communication and satisfaction within their current relationship. Couples-based educational activities and relationship education can be effective in improving communication between couples, relationship quality as well as relationship satisfaction [82–84], and can subsequently be associated with reduced risk of IPV [85, 86]. Social components which draw on activities to improve relationship dynamics utilised in economic interventions have shown promising results for improving partner support in previously violent households, as well as increasing women's autonomy [53]. Relatedly, when interventions exclude couples who wish to stay together (sometimes because of organisational beliefs and ideologies), women who may otherwise seek help may be deterred from doing so and remain at risk.

All of this provides further support for the need for interventions to be developed alongside the community for whom they are intended through the use of participatory approaches, and engaging with local organisations that have invaluable knowledge about the local population, community norms, and beliefs. This should be balanced, of course, with an awareness of the potential for some organisations or community leaders to perpetuate structural violence.

**Future considerations.**   With regard to specific groups of mechanisms, interventions, or outcomes that require more attention in future studies, five main areas stand out: mechanisms relating to resource availability, the existence of harmful stereotypes, interventions to address alcohol and substance abuse, mechanisms leading to successful or harmful couples interventions, and a focus on outcomes that indicate healing and thriving beyond IPV.

**Resources.**   The availability of resources at both individual and institutional levels is central to intervention effectiveness. It is often noted that resource-constrained settings face multiple barriers to effective intervention implementation and delivery [87, 88], but mechanisms around the opportunities and challenges of delivering IPV interventions in resource-constrained settings are still relatively underdeveloped. Future studies should explore the

mechanisms of IPV intervention design and delivery, specifically in resource-constrained settings, to gain a better understanding of how and why barriers to certain interventions exist.

**Stereotypes.** The implications of harmful stereotypes and implicit bias in the provision of care are increasingly being researched [89–93], and this discussion should extend to the provision of services for survivors of IPV. Harmful stereotypes about certain groups of survivors emerged in our review as a potential barrier to culturally sensitive interventions. Qualitative studies are increasingly showing that survivors from certain religious, ethnic, or cultural groups may face barriers to seeking help due to stigma and stereotypes [94, 95]. Future studies should aim to explore the existence of a range of potentially harmful stereotypes, and the impact of provider and institutional prejudice in the provision of services for survivors of IPV, in order to reduce barriers to care and improve IPV response for all survivors. This would be beneficial not only for heterosexual cis-women survivors of violence, but also transwomen and those in same-sex relationships.

**Alcohol and substance abuse.** Although there is a clear link between IPV and alcohol use by both perpetrators and survivors [96–98], IPV interventions addressing alcohol dependence were not prominent among included reviews. In addition to reducing IPV [20, 99], interventions that target substance use can also be important for women's mental health [100]. Future detailed realist evaluations of integrated alcohol and IPV interventions are necessary to understand the mechanisms through which addressing alcohol use in perpetrators and survivors can reduce the risk of IPV and improve mental health.

**Couples interventions.** Caution has been raised about the use of couples therapy in lower- and middle-income contexts [101], and where IPV is characterised by coercive control rather than situational violence [71]. In these cases, it is suggested that couples therapy could lead to both iatrogenic harm and increased risk of violence from the perpetrator [101]. Our review found that empirical detail is currently lacking for mechanisms describing why couples interventions may not be successful or may result in unintended harms. Further research on these mechanisms should be undertaken given the significant role improving relationship dynamics plays in the recovery of women who do not wish to leave their partners.

**Healing and thriving beyond IPV.** Interventions for IPV survivors rarely focus on thriving or positive mental health, instead selecting outcomes indicating reductions in stress or symptoms of poor mental health. Future interventions seeking might usefully incorporate mechanisms identified in this review that address alterations to affective states and cognitions, given the importance of self-kindness and self-compassion in the meaning-making processes necessary for regaining a positive sense of meaning in life [102]. An emphasis on thriving likely requires follow-up that extends beyond the usual 6 or 12 months, but a focus on living well beyond IPV rather than simply surviving could help programmes to create opportunities for change that are truly meaningful for women.

## Limitations

Whilst our use of reviews rather than primary studies allowed us to capture the breadth of interventions and underlying mechanisms in a vast literature base spanning diverse disciplines, one limitation is that some nuance surrounding the mechanisms may have been missed. The necessity for strict inclusion criteria for many systematic reviews and meta-analyses and the lack of focus on grey literature could have excluded some important studies that provide detailed explanations of resources and reasonings, particularly how these work in contexts beyond those captured at the review level.

Our use of reviews may also have missed some newer therapies and approaches that have not yet been formally evaluated or published within primary studies. The search for reviews of

interventions was limited to those addressing intimate partner violence and may have missed some useful studies from other populations such as survivors of sexual assault and torture.

Our findings were strengthened through the triangulation workshops that we held with organisations providing services for survivors of IPV, but both are based in India. Additional triangulation with providers in different settings would be helpful. In this respect, our review contributes to a deeper understanding of how and why certain intervention components work, but is more limited in its discussion of for whom and in what circumstances they are appropriate. Such contextual factors were interwoven throughout our analysis and there are a number of instances in which we explored them further in relation to specific mechanisms. However, analysis at the review level inevitably omitted contextual details that may have emerged more clearly through the use of primary studies. This warrants further investigation into the mechanisms to understand in more depth how they are affected by different settings and populations.

## Conclusion

Supporting the mental health of survivors of IPV is most effectively achieved through holistic, trauma-informed, and individualised interventions that are grounded in their culture and context, that honour the complexity of each woman's situation, and that seek to provide resources to address all of her concerns. Researchers, providers, and organisations seeking to develop interventions to address the mental health of survivors of IPV should consider the design and delivery elements of their programme alongside specific therapeutic components and core components to address basic needs. In addition to improvements in coping and the use of therapeutic techniques, to heal and thrive survivors need an increase in feelings of safety, support and connection, a reduction in environmental stressors, and reductions in violence.

## Supporting information

**S1 File. Review articles used in exploratory scoping.**
(DOCX)

**S2 File. Conceptual model developed after scoping and prior to full review.**
(DOCX)

**S3 File. Final reviews included in the realist synthesis.**
(DOCX)

**S4 File. CMO tables for mechanisms detailed in the review.**
(DOCX)

## Author Contributions

**Conceptualization:** Sharli Anne Paphitis, Laura Asher, Sian Oram.

**Data curation:** Sharli Anne Paphitis, Abigail Bentley.

**Formal analysis:** Sharli Anne Paphitis, Abigail Bentley.

**Investigation:** Sharli Anne Paphitis, Abigail Bentley.

**Methodology:** Sharli Anne Paphitis, Abigail Bentley.

**Project administration:** Sharli Anne Paphitis, Abigail Bentley, Sian Oram.

**Supervision:** Sharli Anne Paphitis, Sian Oram.

**Validation:** Sharli Anne Paphitis, Abigail Bentley, Laura Asher.

**Visualization:** Sharli Anne Paphitis, Abigail Bentley, David Osrin.

**Writing – original draft:** Sharli Anne Paphitis, Abigail Bentley, David Osrin.

**Writing – review & editing:** Sharli Anne Paphitis, Abigail Bentley, Sian Oram.

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
