## [Decision Letter · Decision Letter 0]

18 Feb 2022

Improving the mental health of women intimate partner violence survivors: findings from a realist review of psychosocial interventions

PONE-D-21-29127

Dear Dr. Paphitis,

We’re pleased to inform you that your manuscript has been judged scientifically suitable for publication and will be formally accepted for publication once it meets all outstanding technical requirements.

Kind regards,

Michelle L. Munro-Kramer, PhD, CNM, FNP-BC

Academic Editor

PLOS ONE

Additional Editor Comments (optional):

Excellent work. It is a pleasure to accept this important piece.

Reviewers' comments:

Reviewer's Responses to Questions

**Comments to the Author**

1. Is the manuscript technically sound, and do the data support the conclusions?

Reviewer #1: Yes

Reviewer #2: Yes

2. Has the statistical analysis been performed appropriately and rigorously? 

Reviewer #1: Yes

Reviewer #2: N/A

3. Have the authors made all data underlying the findings in their manuscript fully available?

Reviewer #1: Yes

Reviewer #2: Yes

4. Is the manuscript presented in an intelligible fashion and written in standard English?

Reviewer #1: Yes

Reviewer #2: Yes

5. Review Comments to the Author

Reviewer #1: Thank you for the opportunity to review such important work. The authors have conducted an exceptional review of elements of effective interventions for survivors of intimate partner violence. The manuscripts consolidates much of what we know to be useful to survivors and will be a great resource for practitioners and those who offer services to survivors.

Reviewer #2: I appreciate the opportunity to review this manuscript entitled Improving the mental health of women intimate partner violence survivors: findings from a realist review of psychosocial interventions. Using a realist perspective to conduct a “review of reviews” examining the theories and mechanisms of IPV interventions to improve mental health resulted in a potentially very useful manuscript. Systematically providing both a conceptual model of mechanisms and components and a snapshot of how existing programs fit within that model provides a helpful tool for researchers in understanding the state of the science. This allows the authors to highlight important gaps, such as the ways that highly vulnerable women may lack access to such interventions. I am concerned, however, that the authors overreach in describing these mechanisms. This analysis provides no evidence for the actual effectiveness/efficacy of the various mental health outcomes nor can the authors prove that the described mechanisms are the cause of the outcomes. It would be useful for the authors to acknowledge this limitation and to soften their language in this area.

6. PLOS authors have the option to publish the peer review history of their article (what does this mean?). If published, this will include your full peer review and any attached files.

Reviewer #1: No

Reviewer #2: No

---

## [Editor Report · Acceptance letter]

8 Mar 2022

PONE-D-21-29127 

Improving the mental health of women intimate partner violence survivors: findings from a realist review of psychosocial interventions 

Dear Dr. Paphitis:

I'm pleased to inform you that your manuscript has been deemed suitable for publication in PLOS ONE. Congratulations! Your manuscript is now with our production department. 

Kind regards, 

on behalf of

Dr. Michelle L. Munro-Kramer 

Academic Editor

PLOS ONE